# Did you see it? A Python tool for psychophysical assessment of the human blind spot

**Xiao Ling, Edward H. Silson, Robert D. McIntosh***

School of Philosophy, Psychology and Language Sciences, University of Edinburgh, Edinburgh, United Kingdom

* r.d.mcintosh@ed.ac.uk

## Abstract

The blind spot is a region in the temporal monocular visual field in humans, which corresponds to a physiological scotoma within the nasal hemi-retina. This region has no photoreceptors, so is insensitive to visual stimulation. There is no corresponding perceptual scotoma because the visual stimulation is "filled-in" by the visual system. Investigations of visual perception in and around the blind spot allow us to investigate this filling-in process. However, because the location and size of the blind spot are individually variable, experimenters must first map the blind spot in every observer. We present an open-source tool, which runs in Psychopy software, to estimate the location and size of the blind spot psychophysically. The tool will ideally be used with an Eyelink eye-tracker (SR Research), but it can also run in standalone mode. Here, we explain the rationale for the tool and demonstrate its validity in normally-sighted observers. We develop a detailed map of the blind spot in one observer. Then, in a group of 12 observers, we propose a more efficient, pragmatic method to define a "safe zone" within the blind spot, for which the experimenter can be fully confident that visual stimuli will not be seen. Links are provided to this open-source tool and a user manual.

**Data Availability Statement:** The blind spot mapping tool, its user manual for researchers, and the full data reported in this manuscript, are available at https://github.com/LxliNaGo/BlindSpotMapping.

## Introduction

The blind spot refers to a region in the temporal monocular visual field that corresponds to the optic disc in the nasal hemi-retina, through which the optic nerve leaves the eye [1]. The optic disc is a photoreceptor-free region that encodes no visual information, so it is a physiological scotoma, causing a blind spot in the perceptual visual field. The blind spot is typically centered about 16° temporal to the foveola, 2° below the horizontal meridian, and is roughly an upright oval, with dimensions of around 6° × 7° (Table 1).

This blind region is surprisingly large, but people do not experience a hole in their vision. In binocular conditions, the missing information is compensated by visual inputs from the fellow eye, while in monocular vision, it is believed to be "filled-in" or "completed" with information inferred from remote or global retinal inputs [2–5]. Physiological and psychological

**Funding:** The authors received no specific funding for this work.

**Competing interests:** The authors have declared that no competing interests exist.

evidence indicates that perceptual completion at the blind spot is an active inferential process, rather than a failure to notice that information is missing [6–24].

Given the special "filled-in" status of monocular perception in the blind spot, investigations of the blind spot have potentially great benefits to our understanding of information processing in the visual system. However, a starting point for any such study is the accurate assessment of the blind spot location and size in individual observers. Researchers have applied three general strategies: detection of border points, filling the blind spot region with an adjustable probe object, and perimetric mapping of visual sensitivity across the relevant part of the monocular visual field (Table 1).

### Detection of border points

Detection of border points uses a dot or point visual stimulus to probe the borders of the blind spot. It is the most popular strategy used in blind spot research [9, 25–43]. The typical method is for the observer or experimenter to move a small dot stimulus forth and back across the blind spot borders, and for the observer to report when it perceptually "disappears" (into the blind spot), and when it "appears" (from out of the blind spot). Given multiple estimates for multiple border points, the blind spot border can be mapped in more or less detail, and the location of the blind spot can be estimated from cardinal border points.

### Filling the blind spot

A second strategy is to adjust the location and size of a visual object until it fills the blind spot to the greatest extent possible [2, 7, 19, 38, 44–47]. The typical method is for a moveable and scalable circular object to be used. The initial size of this object is much smaller than the blind spot, so when it is fully within the blind spot, it is invisible to the observer. By adjusting its location and size, the experimenter or the observer can seek to identify the largest unseen size and the specific location at which this size is unseen. The size and location of this maximal probe object provide an estimate of the blind spot, although the estimation accuracy is not high. For instance, when using circular objects, at least one dimension of the blind spot will be underestimated because the blind spot is better approximated by an ellipse than by a circle (note that in Table 1, the width and height given by this strategy are the same because a circular probe object was used).

### Mapping visual sensitivity

The gold standard strategy for mapping the blind spot should be the perimetric mapping of visual sensitivity across the relevant part of the monocular visual field [29, 48–50]. The typical method is that a small dot is displayed many times at different locations, and the observer is asked to report whether or not the dot can be seen. Visual sensitivity for this stimulus can be estimated across repeated trials at each location, and areas having scores lower than a predetermined threshold are taken to indicate the blind spot. Data obtained via this method are of clear psychometric meaning, and can be visualized as a "heat map". However, achieving a high spatial resolution inevitably requires extensive testing. Detailed psychometric mapping is not generally practical for multi-participant experiments, or if the main aim is not to map the blind spot in detail, but simply to define a safe zone within which the experimenter may place stimuli with high confidence that they will not be detected by the observer.

Although many researchers have described their methods in sufficient detail that they could be replicated in an appropriately equipped laboratory, there is to our knowledge no freely available software tool for mapping the blind spot. To solve this problem, we have developed a blind spot assessment tool, written in Python 3 and first run on Psychopy 3.2.3 [51] on

**Table 1. Strategies to map the blind spot and data for healthy human observers in previous studies.**

| First author, Year | Method, Eye | Width | Height | Horizontal Location | Vertical Location |
|---|---|---|---|---|---|
| Berens, 1923 | DBP, R | N, 5.8 | N, 8.8 | NA | NA |
| | | 5.0 | 6.2 | NA | NA |
| Wolf, 1962 | DBP | NA | NA | NA | NA |
| Armaly, 1969 | DBP, R | 7.9 | 10.8 | NA | NA |
| | | 10.2 | 14.9 | NA | NA |
| | | 7.0 | 10.0 | NA | NA |
| | | 9.4 | 13.4 | NA | NA |
| Safran, 1993a | DBP, R | 7.7 | 9.7 | 16.0 | -1.8 |
| | | 7.3 | 9.4 | 16.1 | -2.2 |
| | | 7.1 | 8.5 | 16.0 | -2.1 |
| | | 6.8 | 8.3 | 15.8 | -2.1 |
| | | 6.5 | 8.2 | 15.8 | -2.1 |
| | | 5.9 | 7.1 | 15.6 | -2.1 |
| | | 5.7 | 6.6 | 15.5 | -2.0 |
| Safran, 1993b | DBP, R | 5.9 | 7.1 | 15.6 | -2.1 |
| Murakami, 1995 | FBS | NA | NA | NA | NA |
| Tripathy, 1995 | MVS | NA | NA | NA | NA |
| Awater, 2005 | FBS, R | 2.8 | 2.8 | 15.8 | -1.1 |
| Dolderer, 2006 | DBP | NA | NA | NA | NA |
| Spillmann, 2006 | DBP | NA | NA | NA | NA |
| Araragi, 2008 | DBP | NA | NA | NA | NA |
| Maus, 2008 | DBP, R | 4.7 | 6.0 | 15.0 | -0.5 |
| Araragi, 2009 | DBP, R | 6.3 | 6.4 | 15.8 | -1.4 |
| Dilks, 2009 | DBP | NA | NA | NA | NA |
| Abadi, 2011 | MVS | NA | NA | NA | NA |
| Araragi, 2011 | DBP | NA | NA | NA | NA |
| Baek, 2012 | DBP, R | 5.8 | 6.6 | 14.7 | NA |
| Li, 2014 | DBP, R | 7.6 | 8.3 | NA | NA |
| Ehinger, 2015 | FBS, R | 5.0 | 5.0 | 15.7 | NA |
| Miyamoto, 2015 | FBS | NA | NA | NA | NA |
| Miller, 2015 | DBP | NA | NA | NA | NA |
| Maus, 2016 | DBP, N | 4.1 | 6.5 | 15.0 | NA |
| Ehinger, 2017 | FBS, R | 4.9 | 4.9 | 15.9 | NA |
| Qian, 2017 | FBS, L | 2.9 | 2.9 | 11.8 | -1.6 |
| Wang, 2017 | MVS | NA | NA | 14.3 | -2.1 |
| Chen, 2017 | DBP, N | 5.4 | 6.6 | 15.4 | -1.8 |
| Saito, 2018 | FBS | NA | NA | NA | NA |
| Revina, 2020 | DBP, N | 5.2 | 6.2 | 16.4 | NA |

Data are expressed as means in degrees of visual angle. Locations are eccentricities relative to the fixation point. Data obtained from multiple experimental settings within one study are expressed as multiple data entries. When data are available for both eyes, only the right blind spot data are shown, otherwise they are labeled with "L" for left blind spot, or "N" for "not clear". "NA" referes to "not available". DBP = Detection of Border Points, FBS = Filling the Blind Spot, MVS = Mapping Visual Sensitivity. The table is based on a literature search conducted on Web of Science, searching papers in English, published until 2019, with keywords of "blind spot" and "filling-in" or "completion". Studies that detailed their blind spot mapping methods for humans were included and demonstrated in the table.

Windows 10 and 7 operating systems. This tool provides estimates of blind spot location and size via a stepwise testing that has three parts: a "Border points detection section" that estimates border points, a "Staircase section" that adjusts the border point estimates, and a "Validation section" that validates estimation results. In the "Validation section", we also provide the option of perimetric mapping of the blind spot. This tool can be run with/without eye-trackers (tested with Eyelink 1000). Experimental parameters can be adjusted in the graphical user interface or by editing configuration files and source codes.

Here we describe the main methods implemented by this tool. In Experiment 1, we illustrate the perimetric mapping method by generating a detailed heat map of the blind spot for one observer. These heat map data also serve as a preliminary validation of the much shorter, stepwise method for blind spot estimation. In Experiment 2, we validate this stepwise method in 12 further observers. Finally, based on our data, we recommend a maximally efficient mapping protocol to estimate the location and size of the blind spot, and we specify some simple rules to enable researchers to confine visual stimuli safely within the blind spot during experimental investigations.

## Description of the blind spot assessment tool

This tool uses psychophysical methods to estimate the location and size (width and height) of the blind spot, using the simplifying assumption that the blind spot is an upright ellipse that is symmetric around the horizontal and vertical axes, given the shape of the optic disc [52]. A full-length mapping block has three separable, stepwise experimental sections: Border points detection section, Staircase section, and Validation section (the perimetric, gold standard procedure is an option for the Validation section). Here we describe the procedure(s) for each of these sections and our settings for them, as applied in our experiments. Note that all the values described below represent the default settings of the tool, but they can be adjusted by the user.

### Apparatus settings

Experiments were performed in a dark room, with a 21-inch CRT monitor (G225f, Viewsonic) measuring 40 cm (horizontal) by 30 cm (vertical). Its resolution was set to $1280 \times 960$ pix, with a refresh rate of 85 Hz, and the actual luminance was 27 cd/m$^2$ when displaying mid-level gray (RGB value 128, 128, 128) on full screen. The monitor gamma value was calibrated before experiments. The observer's head was fixed on a chinrest centrally in front of the screen at a viewing distance of 57 cm to the screen center. At this distance, 1 cm in the central area of the screen subtends 1˚ (around 32 pix), so that the screen subtended 40˚ (horizontal) and 30˚ (vertical) of visual angle.

The assessment of the blind spot requires monocular viewing. The tool can assess the blind spot in either eye, but the right eye was the test eye in our experiments, and the left eye was covered with an eye patch or glasses cover. To improve data accuracy, the tool is intended to interface with an eye-tracker (Eyelink 1000, SR Research), but it can also run as a standalone tool (eye-tracker-off mode). Psychopy also supports other eye-trackers (GazePoint, SR Research, and Tobii), but we have only tested and used the tool with Eyelink 1000. The native nine-point (center, upper, lower, left, right, and four corners) calibration procedure of Eyelink 1000 was used, which calibrates the eye-tracker in both horizontal and vertical dimensions.

Throughout all experimental blocks, the eye movement tolerance was set to 1.5˚. When the observer's fixation was recorded as more than 1.5˚ away from the center of the fixation object, the color of the fixation object changed, responses were ignored and the current trial would re-run until successful. With the tool in eye-tracker-on mode, mouse clicks and key presses will be used to assess the blind spot only when fixation is within required limits, but all eye-

tracking data and observer/experimenter inputs will be logged. Users can modify the configuration files to save these additional data if they want.

This fixation tolerance (1.5˚), which is larger than the likely deviation of true fixation positions, was chosen to allow for a small amount of eyetracker signal drift during extended blocks of trials, without triggering regular fixation errors, which would be frustrating and time-consuming. The default tolerance level provides a balance between precision and efficiency, but a lower tolerance threshold can be set by the user if highly precise blind spot estimation is required. However, if using a lower fixation tolerance, we would recommend regular drift correction during blocks of trials, as described in the user manual ("eyetrackingconfig.json" item of Section 3.3).

### Fixation object

In both experiments, the fixation object was a bull's eye, with an outer circular annulus surrounding an inner circular spot, located at (-300, 0) pix, equivalent to around 9.34˚ left to the screen center ((0, 0) pix). The outer circle diameter was 40 pix (1.24˚), and the inner circle diameter was 12 pix (0.38˚). When the observer was within the 1.5˚ tolerance limit, the outer circle was white and the inner circle was black. When fixation was detected outside this limit, the colors changed to blue and red, providing a visible error signal to the observer.

### Border points detection section

This section applies a three-step method to detect border points and then estimates the location of the blind spot based on cardinal border points (Fig 1). This method is based on the simplifying assumptions that the blind spot can be approximated as an upright ellipse which is symmetrical in horizontal and vertical dimensions, and that the location of the blind spot is the center of this ellipse.

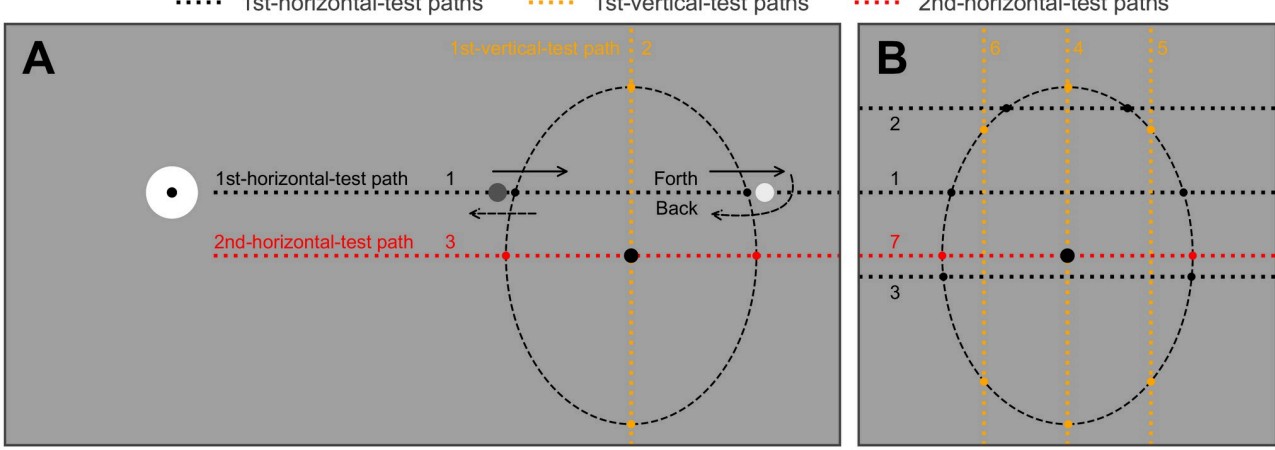

**Fig 1. Two variants of the three-step blind spot estimation method.** The bull's eye is the fixation object, and the dashed ellipse is the right blind spot to be tested. Numbers indicate the order of paths being run. (A) The standard procedure, which is used in "Low" mode of this tool. The observer moves a flickering probe across the blind spot along the 1st-horizontal-test path (path 1), and reports when the probe perceptually "disappears" and "reappears" at each edge of the blind spot. The observer is then asked to do the same task along the same path but in the opposite direction. These forth and back trials form one round trip. The experimenter should determine the number of round trips for each path (three in our experiments). After completing the 1st-horizontal-test, the same tasks are performed in 1st-vertical-test (path 2) and 2nd-horizontal-test (path 3). The blind spot location is determined by the intersection of paths 2 and 3, while the blind spot width and height are determined by paths 3 and 2. (B) A variant that detects 14 border points. Since more border points are detected, the estimated contour will be closer to the true blind spot shape. This variant is applied in "Medium" mode of the tool. The same tasks are performed in 7 paths and the order is suggested by numbers. Full descriptions about "Low", "Medium" and "High" are provided in the user manual ("Mode" item of Section 3.3).

The observer is asked to fixate the fixation object using the test eye throughout this section. A flickering circular probe object of 12 pix diameter (0.38°), flickering between white and black at 20 Hz, is under the continuous control of the observer by computer mouse. The observer is asked to move the flickering probe forth and back across the blind spot, during which the probe perceptually "disappears" on entering the blind spot and "reappears" on leaving the blind spot. The observer is asked to report where the probe disappeared and reappeared by clicking the mouse, so that in one round trip (i.e., a forward trial plus a backward trial), the inner and outer border points are each reported twice, once moving inward and once moving outward. These clicks are then clustered into two subgroups and separately averaged as the estimates of two border points. Ideally, these estimates should be on the 50% visibility boundary (50% likelihood of reporting "Yes I see it"). In this section, the experimenter should monitor the observer and stop the observer after a predetermined number of round trips has been completed (three in our experiments, i.e., three reports for each direction for each of two border points). Note that the experimenter may allow the observer to continue for more (or fewer) rounds, as desired.

As depicted in Fig 1A, this moving-and-clicking task should be performed in horizontal (1st-horizontal-test phase), vertical (1st-vertical-test) and horizontal direction again (2nd-horizontal-test) to detect multiple border points and estimate the location of the blind spot based on the four cardinal border points (i.e., the three-step method, see Baek et al., 2012). This standard three-step procedure, illustrated in Fig 1A, is designated as the "Low" mode, which aims to estimate the blind spot location and size but not its shape. The "Low" mode of this section estimates six border points, among which the four estimated from the 1st-vertical-test and 2nd-horizontal-test are the cardinal border points (for height and width, respectively, and to estimate the location), while the two estimated from the 1st-horizontal-test are used only to define the 1st-vertical-test phase. Fig 1B shows a variant procedure for the "Medium" mode, which detects fourteen border points. The "High" mode, which detects 22 border points, is also available as an option, but is not shown in Fig 1, and was not tested in our experiments.

## Staircase section

The "Low", "Medium", and "High" modes are for the "Border points detection section" only, but they also impact the behavior of the "Staircase section". For the "Low" mode, four cardinal border points out of all six points, which were estimated by the 1st-vertical-test and the 2nd-horizontal-test, will then be examined in the Staircase section. This "Low" mode design allows for the estimation of width and height, with no further details of shape. For "Medium" and "High" modes, all border points (14 and 22, respectively) will be tested in the Staircase section, so that all points can be used to refine the estimated blind spot shape, in addition to the blind spot size.

This section has many trials. As depicted in Fig 2A, in each trial, the fixation object is displayed on the screen, and the program is waiting for the observer to press the spacebar to trigger the target object by pressing the spacebar. Then a flickering target object, identical to the probe object used in the Border points detection section, is presented on the screen for a short duration (400 ms). After the offset of the target object, the screen is cleared, a computer-synthesized man's voice sound "Did you see it?" is played, and the observer needs to indicate whether or not the target object was seen, by keypress ("left arrow" for "no" or "right arrow" for "yes").

The target object is displayed along the straight line that connects the corresponding border point and the estimated blind spot location. The target object will be farther from the location in the current trial if a "no" response (i.e., target object not being seen) was given in the last

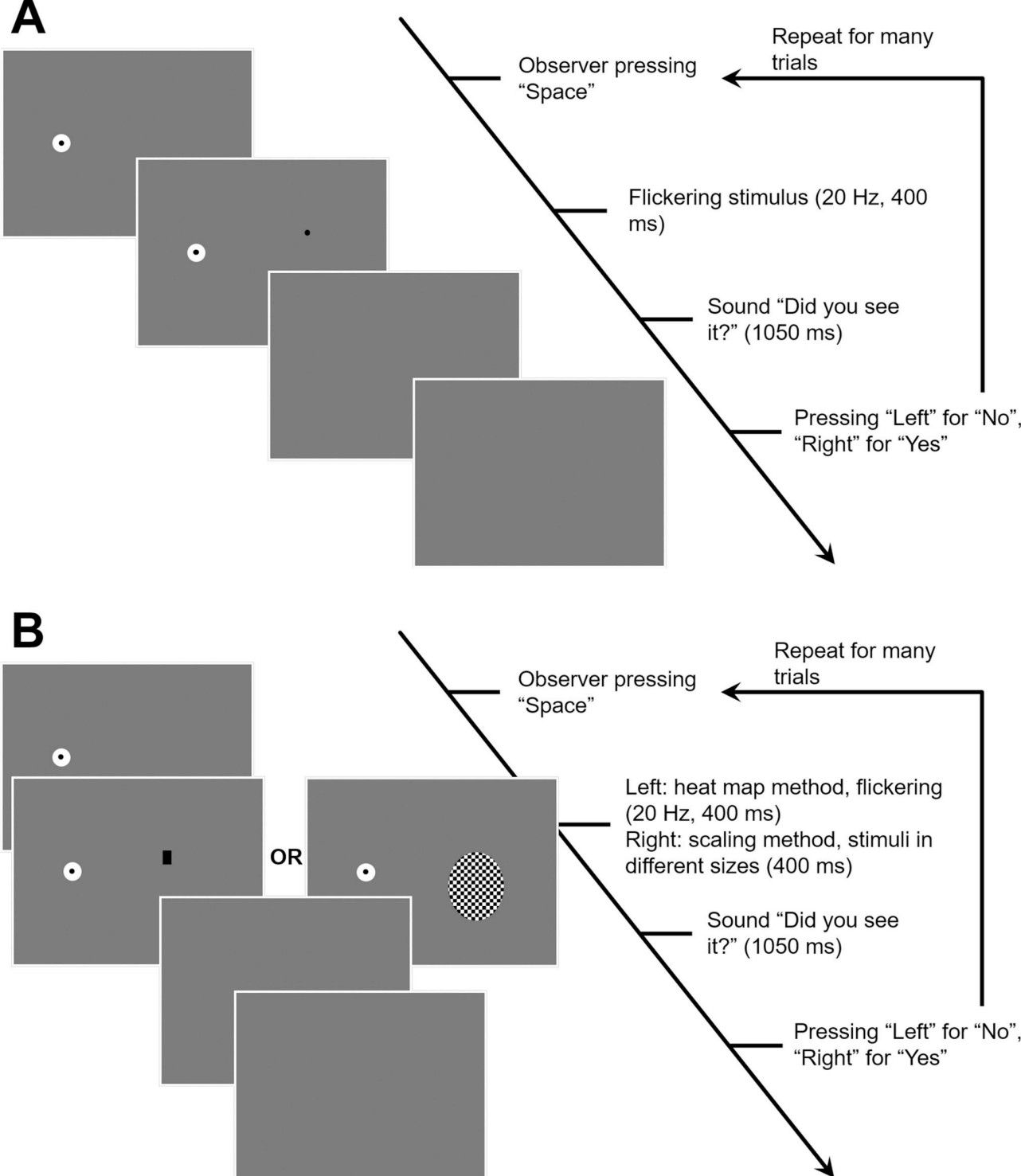

**Fig 2. Schematic for staircase section and validation section.** (A) The Staircase section. (B) The Validation section. Note that there are two different options, "Heat Map" and "Scaling". See text for details.

trial, or closer to the location if a "yes" response was given in the last trial. The initial target object positions are the detected border points, and subsequent positions will be calculated by the initial position subtracting the increment (a positive increment indicates a position closer to the blind spot location). The initial increment is predetermined, while subsequent increment values will be calculated based on the predetermined step sizes, the observer's responses, and down-up rules, as follows.

The staircase rules are: initial increment of 50 pix, step sizes of [3, 7, 15, 30] pix, 20 successful staircase trials (in eye-tracker-on mode, trials with unstable fixation will be ignored and re-run), 1-up-3-down, 4-reversals, and final-1-trial for computation of the threshold value (see Fig 3 for an example). The up-down rule determines the "visibility" of the examined border points, and a 3-down-1-up rule should target the 79% correct rate [53, 54]. So, after adjustment by the Staircase section, the border points should be 79% "invisible" (79% probability answering "no"), in other words, 21% visible. In this way, the Staircase section adjusts the initial border point estimates such that they have known psychometric properties.

Once original border points have been examined, the adjusted estimates will be the final estimated border points. The width and height of the blind spot are calculated accordingly.

## Validation section

The Validation section has two options of procedures. The first is "Heat Map", which is an extensive mapping of visual sensitivity (Fig 2B left). This procedure should be the gold standard for mapping the blind spot. It tests the observer's visual sensitivity across a rectangular visual field that fully covers the estimated blind spot (1.1 times the estimated width and 1.2 times the estimated height). It evenly grids this rectangular visual field into user-specified numbers of rows and columns (by default 10 × 10), creating a grid of rectangular cells. The visual stimuli are target objects fully filling a single cell, and flickering between black and white at 20 Hz. These target objects are presented one-by-one, 10 times per cell (400 ms per trial), in a randomly shuffled order.

This procedure has many trials. As depicted in Fig 2B, in each trial, the fixation object is displayed on the screen, and the program is waiting for the observer to press the spacebar to trigger the onset of the target object. After the offset of the target object, the screen is cleared, the computer-synthesized man's voice sound "Did you see it?" is played, and the observer needs to report whether or not the target object was perceived ("left arrow" for "no", or "right arrow" for "yes"). The visual sensitivity for the target object can be calculated across the 10 trials at each cell, resulting in a visual sensitivity matrix, or "heat map". The region within which the visibility score is below a chosen arbitrary threshold (such as 0.5) can be regarded as the blind spot. In this way, "visibility" is quantified as a proportional likelihood of the target being seen, so that the operational definition of the blind spot area has a clear psychophysical meaning.

The default validation option is "Scaling", which is a much quicker but much more approximate procedure than the option of "Heat Map" (Fig 2B right). This procedure uses elliptical probes that are symmetrical around the horizontal and vertical axes, with the same aspect ratio (height/width) as the estimated blind spot width and height, but varying scaling ratios relative to the estimated blind spot size (10 uniformly distributed scaling coefficients in the range of [0.6, 1.2]). Each tested size of the elliptical target object is presented at the estimated blind spot location for a short duration (400 ms) in each of several trials (10 trials per size).

In each trial, the fixation object is displayed on the screen, and the program is waiting for the observer to press the spacebar to trigger the onset of the target object. After the offset of the target object, the screen is cleared and the computer-synthesized man's voice sound "Did you see it?" is played. The observer needs to report whether or not the target object was perceived

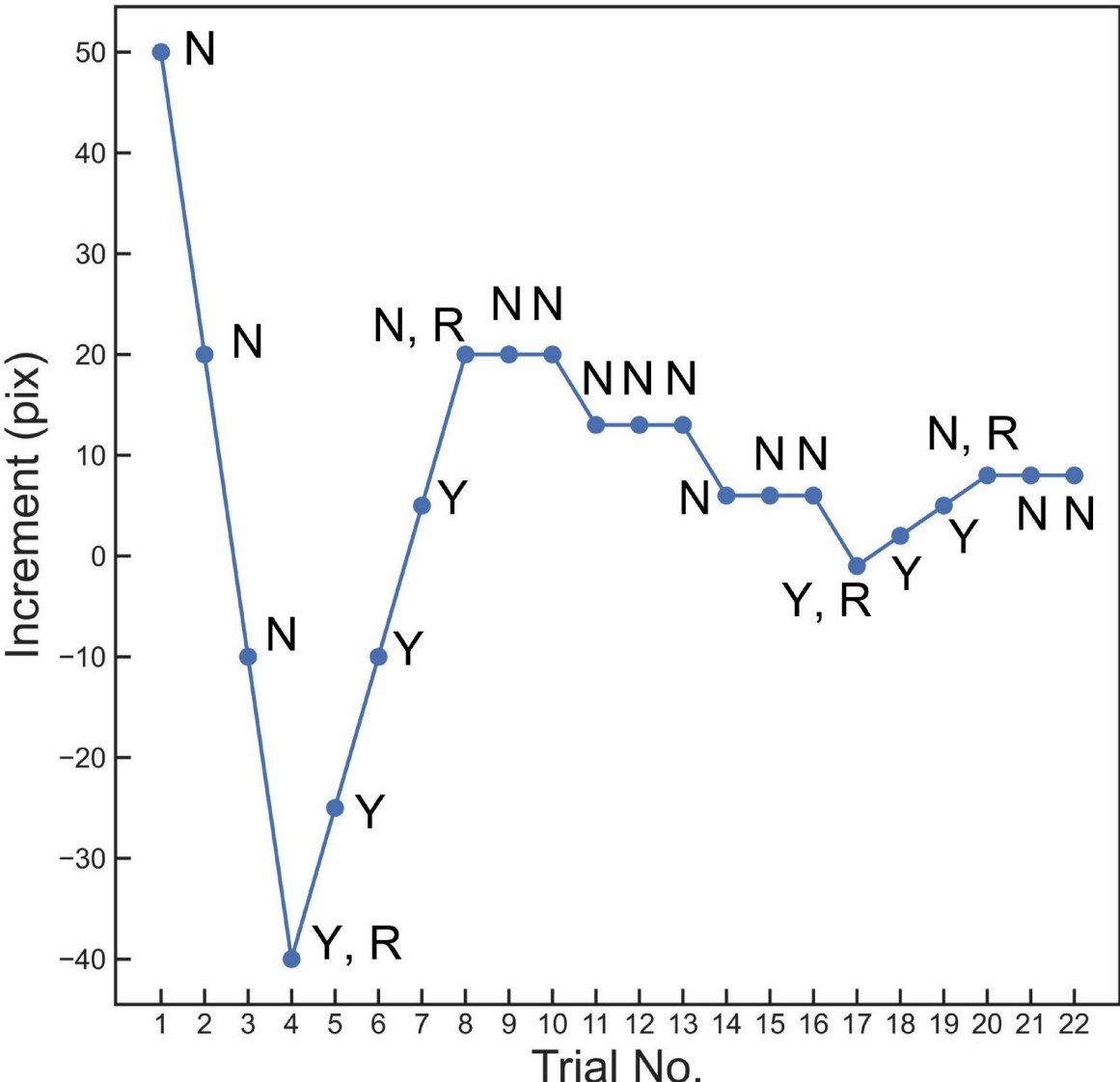

**Fig 3. Staircase trace example.** An example of staircase trace reflecting default staircase rules (initial increment = 50 pix, increment step sizes = [30, 15, 7, 3], 1-up-3-down, 4-reversals, 20 trials). This trace depicts changes in the increment for one single border point examined by the staircase section. Each data point is labeled with the observer's response (N = "No", Y = "Yes"), and reversals are marked with R. Note that 0 increment represents the original position of the estimated border point, and that the up-down rule remains 1-up-1-down before the first reversal. Before the first reversal has been made, any answer will change the increment by one step size, while after the first reversal, one positive answer ("Y") will increase the increment, but every three negative answers ("N") will decrease the increment by one step size.

("left arrow" for "no", or "right arrow" for "yes"). In this way, the "visibility" for each size is quantified as a proportional likelihood of the target being seen.

## Methods

Both experiments used the same equipment and parameters, as described above, unless otherwise indicated.

## Experiment 1

Experiment 1 was an extensive investigation of the right-eye blind spot in a single observer (first author XL, a 24-year-old male), conducted to validate the three-step procedure used in the "Border points detection section" and the staircase procedure in the "Staircase section" against the gold-standard of the "Heat Map" procedure in the "Validation section".

The observer performed one session (one visit to the lab), which had three blocks. The first block consisted of a "Low" mode "Border points detection section", a "Staircase section", and a "Validation section" using 15 rows × 15 columns "Heat Map" (so at least 6 [point] * 6 [trial/point] trials for Border points detection, 4 [point] * 20 [trial/point] for Staircase, and 15 * 15 [cell] * 10 [trial/cell] for Validation). The second block had the same sections as the first block, but the "Heat Map" was 10 rows × 10 columns (so at least 36 trails for Border points detection, 80 for Staircase, and 1000 for Validation). The third block had a "Medium" mode "Border points detection section", a "Staircase section", and a "Validation section" using a "Scaling" procedure (so at least 84 trials for Border points detection, 280 for Staircase, and 100 for Validation), but these validation results were ignored and not analyzed. No training block was performed because the observer was already skilled. During each block, there was no break, but between blocks, the observer took a 5 min break. These three blocks were completed within 60, 30, and 20 min, respectively.

Data from Experiment 1 were used to qualitatively compare the estimated blind spot border points, before and after the staircase, against the heat maps.

## Experiment 2

Experiment 2 was conducted to investigate the application of a standard, full-length block in a larger group of observers. Multiple blocks were run for each observer to assess within-subject consistency.

We collected the right blind spot data for 12 (5 men and 7 women, average age = 26.4, age range = [22, 48]) out of 15 observers, after a training block. Three observers were excluded before or during the first session. Two of them were excluded due to the eye-tracker failing in monitoring their eyes, and the other one withdrew from the experiment. This experiment was approved by the Research Ethics Committee of the School of Philosophy, Psychology and Language Sciences, University of Edinburgh. All observers gave written informed consent.

After the training block, each observer performed six blocks within two to three sessions, depending on personal preference. Each block had a "Low" mode "Border points detection section", a "Staircase section", and a "Validation section" using the "Scaling" procedure. Every observer was allowed to take a break at any time during a block and between blocks. Most observers could complete a block within 30 min, and all observers completed a session in no more than one hour.

Data from Experiment 2 were used to analyze the size and location of the blind spot and the precision and accuracy of the estimates.

## Results

### Experiment 1

**Heat map data.**　Two heat maps were generated based on data from separate blocks: 10 × 10 (Fig 4A and 4B) and 15 × 15 (Fig 4C and 4D). In the 10 × 10 heat map, the tested visual field was a rectangular region that ranged from 12.74˚ to 20.00˚ in horizontal (7.26˚ width) and -5.15˚ to 4.01˚ in vertical (9.16˚ height). It was evenly gridded to 100 rectangular cells, and each cell had the same size of 0.73˚ × 0.92˚ (width × height). The percent-positive response

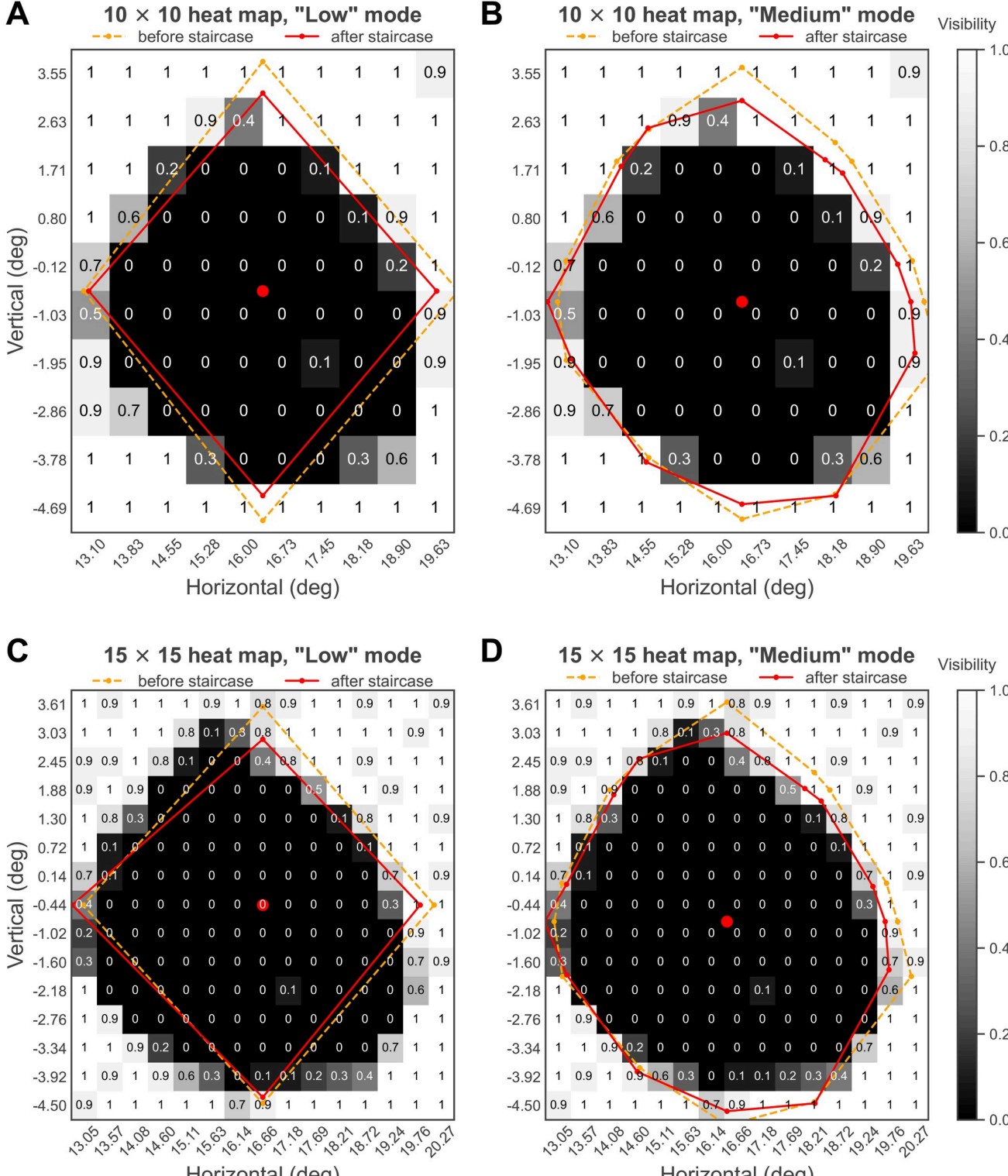

**Fig 4. Heat maps and blind spot location and border points estimated in Experiment 1.** The blind spot border points and the heat map data were from the same experimental block for (A) and (C), but not for (B) and (D). The two heat maps align with each other, and "Medium" mode has captured more shape details of the blind spot. Note that a non-zero (0.1) visibility spot inside the estimated blind spot appeared on both heat maps at the same position (see Discussion).

score (visibility) across the 10 trials in each cell was computed. The 15 × 15 heat map covered a similar area (width = 7.74˚, height = 8.69˚, horizontal range = [12.79˚, 20.53˚], vertical range = [-4.79˚, 3.90˚]), but with a greater number of smaller cells (225 cells, cell width = 0.52˚, cell height = 0.58˚), for a higher spatial resolution.

The blind spot was operationally defined as those regions with a visibility score lower than 0.5. The two heat maps similarly characterized the blind spot as an approximate upright oval. It was generally, though not perfectly, symmetrical in vertical and horizontal dimensions. This symmetry supports the assumption regarding the blind spot shape which is made by the three-step method used in the "Border points detection section".

These two heat maps also suggested that the blind spot region has a sharp boundary. The marginal blind spot cells had mean visibility values of 0.08 (n = 20, SD = 0.12) and 0.13 (n = 33, SD = 0.14) for the 10 × 10 and 15 × 15 heat maps, respectively; the visibility values of marginal normal field cells were 0.9 (n = 24, SD = 0.15) and 0.86 (n = 34, SD = 0.14), respectively. Given that 0.9 was not uncommon even for cells that were apparently outside the blind spot, 0.9 and 0.86 did not suggest any reduction in visual sensitivity around the outer border of the blind spot but might be caused by response errors; 0.08 and 0.13 also did not suggest any subtle visual sensitivity around the inner border of the blind spot, for the same reason.

The blind spot width and height can be estimated by the separations between the left and right, and the upper and lower boundaries of the estimated blind spot area. For example, the height can be estimated by the separation between the upper boundary of the upper blind spot cell and the lower boundary of the lower blind spot cell. The 10 × 10 heat map indicated that the blind spot width and height were around 6.57˚ and 7.36˚, respectively, while the data for the 15 × 15 heat map were 6.76˚ for width and 7.54˚ for height. These data were in line with the estimates of the border points detection and staircase sections.

We chose not to define a blind spot location from the heat maps because the observed blind spot was not a perfectly regular shape. Even if we were to operationalize the location as the intersection between the dimensions of maximum width and height, this would not specify a unique point, because there were multiple points through which the maximum width and height could be estimated. Nonetheless, it is obvious in all four subplots, that the blind spot location estimated from the "Border points detection section" fell within the central region of the blind spot heat map.

The "Medium" block did not estimate a heat map. The blind spot border points estimated in the "Medium" block were overlaid against the two heat maps estimated from the other two blocks for comparison (Fig 4B and 4C). Of course, because fourteen border points were mapped instead of just four, the "Medium" mode revealed more details of the blind spot shape than "Low" mode did. However, the much more efficient "Low" mode may estimate the blind spot size and location adequately for most purposes.

**Blind spot size and location.** The blind spot sizes (calculated based on border points after the Staircase section) estimated by "Low" and "Medium" blocks were close. The estimated width values were around 6.59˚ and 7.03˚ for two "Low" mode blocks (Fig 4A and 4C, respectively), and 6.96˚ for the "Medium" mode block (Fig 4B and 4D), while the height values were 7.54˚, 7.16˚, and 7.56˚, respectively. These data were well within the range of values reported in earlier studies that also used the detection of border points strategy (Table 1).

For the two "Low" blocks, the corresponding blind spot locations were at 16.36˚ horizontally, -0.57˚ vertically (Fig 4A), and 16.66˚ horizontally, -0.44˚ vertically (Fig 4C). For the "Medium" block, the blind spot was located at 16.46˚ temporal to the fixation, and 0.78˚ lower than the horizontal meridian (Fig 4B and 4D). These data aligned well with each other and were well within the range of values reported in earlier studies (Table 1).

It is clear that raw blind spot border points were adjusted toward the blind spot location by the Staircase section, although this shrinking effect was apparently more subtle for the nasal border points. In general, the Staircase section tended to move blind spot border point estimates toward lower visibility levels. This is expected, due to the fact that the Staircase section aimed to determine border points at the 21% visibility level, while before the Staircase section, the raw border points should be at around 50% visibility level.

Very few adjusted border points lay within 0.2 visibility (rounded from 21% visibility, in other words, 79% invisibility) cells. On Fig 4C (15 × 15 heat map, 0.5° resolution), only one point lay in 0.4, while other three points lay in 0.8 to 1.0; on Fig 4D (15 × 15 heat map), only one point lay in 0.3 while thirteen points lay in 0.7 to 1.0. So, based on the heat maps, we failed in confirming that the Staircase section effectively calibrated border points to 21% visibility. This may be explained by the relatively coarse resolution provided even by the smaller cell size (0.5°) in the higher resolution heat map. This resolution may not have been high enough to detect the rather narrow, 0.2 visibility region, particularly given an eye movement tolerance of 1.5°.

## Experiment 2

In this experiment, twelve observers were tested, each of whom completed six blocks in two to three sessions. Each block estimated four cardinal border points before and after the Staircase section (so eight in total), and one blind spot location. Two blocks (the first block for observers No. 2 and 5) were aborted or excluded due to interruption or data-saving problems, so 70 out of 72 blocks were analyzed in total.

**Blind spot location and size.** The adjusted blind spot estimates for each observer are shown in Table 2. On average (bottom row of Table 2), the right blind spot was located at 16.00° temporal (right) to the fixation (SD = 0.52°, Range = [15.17°, 16.73°], n = 12), and slightly lower than the horizontal meridian (mean = -2.05°, SD = 1.21°, Range = [-4.33°, -0.50°], n = 12).

To evaluate the precision of location estimates, the standard distance (Stdist) was calculated for each observer. The Stdist is the root-mean-square deviation of the Euclidean distance of the estimated point from its mean position, and a lower Stdist value indicates higher precision. The formula for Stdist is:

$$Stdist = \sqrt{\frac{1}{n}\sum_{i=1}^{n}[(x_i - \bar{x})^2 + (y_i - \bar{y})^2]} \qquad (1)$$

where x and y refer to horizontal and vertical coordinates, respectively, and n is the number of blocks.

The location Stdist data are listed in Table 2 (mean = 0.19°, SD = 0.07°, Range: [0.09°, 0.29°], n = 12). The location was dependent on border points detected in the "Border points detection section", but the final width and height were estimated based on the border points adjusted by the "Staircase section", where they were adjusted against the estimated location, so the variation in the precision of location estimates might introduce an error into the estimated size (see below for analysis of the precision of width estimation), but it was not substantial. At the individual level, dividing the Stdist by the corresponding minor axis length (the smaller of blind spot width and height) for each observer, the quotient ranged from 0.02 to 0.05. The mean quotient was 0.03, with a SD of 0.01, so this error had a less than 5% influence on size estimates.

The location estimated in different blocks for some observers (observers No. 5, 8, 12, and 13) appeared more inclined to drift vertically than horizontally (Fig 5). Indeed, the variation of

**Table 2. Summary of blind spot estimation results for each observer.**

| Observer | Location (H, V) | Location H Range | Location H SD | Location V Range | Location V SD | Location Stdist | Location Distance Range** |
|---|---|---|---|---|---|---|---|
| 1 | (15.21, -1.51) | [15.00, 15.63] | 0.22 | [-1.94, -1.18] | 0.30 | 0.28 | [0.08, 0.53] |
| 2 | (15.17, -2.19) | [14.99, 15.27] | 0.11 | [-2.31, -2.02] | 0.11 | 0.14 | [0.06, 0.20] |
| 3 | (16.51, -0.52) | [16.39, 16.64] | 0.09 | [-0.69, -0.36] | 0.11 | 0.12 | [0.06, 0.20] |
| 4 | (16.42, -0.50) | [16.23, 16.59] | 0.12 | [-0.69, -0.31] | 0.15 | 0.15 | [0.10, 0.20] |
| 5 | (15.91, -2.40) | [15.72, 16.21] | 0.23 | [-2.86, -1.99] | 0.34 | 0.29 | [0.22, 0.55] |
| 7 | (16.38, -1.40) | [16.25, 16.59] | 0.14 | [-1.57, -1.25] | 0.15 | 0.17 | [0.16, 0.22] |
| 8 | (16.24, -2.01) | [15.98, 16.50] | 0.21 | [-2.36, -1.45] | 0.38 | 0.27 | [0.16, 0.62] |
| 9 | (16.18, -2.27) | [15.92, 16.31] | 0.14 | [-2.46, -2.05] | 0.15 | 0.18 | [0.06, 0.33] |
| 10 | (15.92, -4.33) | [15.71, 16.23] | 0.21 | [-4.49, -4.04] | 0.16 | 0.27 | [0.10, 0.34] |
| 11 | (15.31, -3.03) | [15.19, 15.49] | 0.13 | [-3.33, -2.84] | 0.22 | 0.16 | [0.13, 0.33] |
| 12 | (16.05, -3.66) | [15.84, 16.25] | 0.15 | [-4.45, -3.18] | 0.48 | 0.20 | [0.05, 0.81] |
| 13 | (16.73, -0.74) | [16.60, 16.79] | 0.07 | [-1.21, 0.08] | 0.50 | 0.09 | [0.11, 0.82] |
| Average* | (16.00, -2.05) | [14.99, 16.79] | 0.16 | [-4.49, 0.08] | 0.29 | 0.20 | [0.05, 0.82] |

| Observer | Width | Width Range | Width SD | Height | Height Range | Height SD | Raw Border Points Sum Stdist | Border Points Sum Stdist |
|---|---|---|---|---|---|---|---|---|
| 1 | 5.97 | [5.81, 6.19] | 0.15 | 6.35 | [5.99, 6.61] | 0.26 | 0.97 | 0.87 |
| 2 | 5.79 | [5.43, 5.97] | 0.21 | 6.45 | [6.09, 6.74] | 0.30 | 0.61 | 0.59 |
| 3 | 6.71 | [6.54, 6.85] | 0.11 | 7.57 | [7.36, 7.81] | 0.19 | 0.42 | 0.65 |
| 4 | 7.12 | [6.58, 7.52] | 0.31 | 6.83 | [6.60, 7.05] | 0.20 | 0.74 | 0.85 |
| 5 | 6.72 | [6.44, 7.00] | 0.24 | 5.79 | [5.42, 6.20] | 0.37 | 1.11 | 1.15 |
| 7 | 6.57 | [6.21, 6.71] | 0.18 | 7.40 | [7.12, 7.65] | 0.20 | 0.85 | 0.68 |
| 8 | 5.39 | [4.82, 5.69] | 0.33 | 7.47 | [6.95, 7.76] | 0.34 | 1.03 | 1.01 |
| 9 | 5.92 | [5.73, 6.14] | 0.15 | 5.91 | [5.77, 6.06] | 0.11 | 0.68 | 0.69 |
| 10 | 7.44 | [7.24, 7.66] | 0.15 | 8.14 | [7.85, 8.49] | 0.26 | 1.10 | 0.98 |
| 11 | 6.15 | [6.00, 6.21] | 0.08 | 7.91 | [7.46, 8.28] | 0.36 | 0.75 | 0.72 |
| 12 | 5.88 | [4.97, 6.51] | 0.52 | 7.54 | [7.02, 8.08] | 0.44 | 0.75 | 1.16 |
| 13 | 5.71 | [5.19, 6.27] | 0.38 | 6.94 | [6.41, 7.24] | 0.34 | 0.62 | 0.73 |
| Average* | 6.28 | [4.82, 7.66] | 0.26 | 7.02 | [5.42, 8.49] | 0.30 | 0.83 | 0.86 |

All values are in degrees of visual angle. Stdist refers to the standard distance (square root of the mean of squared Euclidean distance). It quantifies how 2D points spread, so it is a measure of precision. SD refers to standard deviation.

Average*: The averaged SDs are the square root of mean variances, so may differ from the in-text SDs of 12 individual means. The averaged Stdists are the square root of average squared Stdists. The averaged ranges are the Min and Max values of all 70 individual data, so may be different from in-text ranges that are the Min and Max values of 12 individual means.

Location Distance Range**: The range of Euclidean distances between estimated locations and the averaged location.

vertical coordinates of the blind spot center for these observers (vertical SD = 0.34˚, 0.38˚, 0.48˚, and 0.50˚, mean = 0.43˚) was higher than the horizontal variation (horizontal SD = 0.23˚, 0.24˚, 0.15˚, and 0.07˚, mean = 0.17˚). In fact, there was an overall trend that vertical precision was lower than horizontal precision (Table 2, see individual and Average values for horizontal and vertical SDs).

The mean blind spot sizes for all observers are also listed in Table 2. Overall, the blind spot width was 6.28˚ (SD = 0.62˚, CV = 0.10, Range = [5.39˚, 7.44˚], n = 12), and the height was 7.02˚ (SD = 0.77˚, CV = 0.11, Range = [5.79˚, 8.14˚], n = 12). The size data, as expected, were close to previous studies that applied similar three-step methods (Araragi et al., 2009; Baek et al., 2012; Chen et al., 2017; Maus & Whitney, 2016; Safran, Mermillod, et al., 1993; Safran, Mermoud, et al., 1993, also see Table 1).

The precision of size estimates can be evaluated based on the individual SD of width and height. The individual SD of width (mean = 0.23˚, SD = 0.12˚, CV = 0.53, Range = [0.08˚,

0.52˚], n = 12) and that of height (mean = 0.28˚, SD = 0.09˚, CV = 0.31, Range = [0.11˚, 0.44˚], n = 12) are listed in Table 2. The difference between individual SD of width and that of height was not significant (two-tailed independent t test, t = 1.01, p = 0.32, df = 22). But given that the width was generally smaller than the height, the variation was relatively greater for width estimates.

The relatively lower precision in width may be explained by the aforementioned lower precision in vertical location. Note that, in the border points detection section, the width is estimated by the 2nd-horizontal-test, which is determined by the estimated blind spot vertical location (Fig 1A), so variations in vertical location estimation may impact the width estimation.

Similar to Experiment 1, the Staircase section made the estimated blind spot border points more conservative, although this shrinking effect was negligible for observers No. 7 and 11 (Fig 5). This effect was as expected because the estimated border points before the staircase should target around 50% visibility level, while the Staircase section should calibrate them to 21% visibility level.

The overall precision of border points estimates for an observer can be evaluated by summing the standard distance (Stdist) values of all four cardinal border points. The overall precision of raw border points (mean = 0.80˚, SD = 0.21˚, Range = [0.42˚, 1.11˚], n = 12) and that of vertices calibrated by the Staircase section (mean = 0.84˚, SD = 0.19˚, Range = [0.59˚, 1.16˚], n = 12) were not significantly different (two-tailed paired t test, t = 0.81, p = 0.437, df = 11). So the staircase section did not appear to increase overall precision.

**Validation data.** The data obtained in the scaling validation section could estimate a probability of giving positive responses for each elliptical target object. This probability could be taken as an index of the visibility of the corresponding ellipse. Data across all sizes could be fitted with a sigmoid function. The sigmoid function is:

$$f(x) = \frac{1}{1 + e^{-[t(x-x_0)]}} \times 100\% \tag{2}$$

in which t, and x0 are constants. The function ranges from 100% to 0%; that is, from complete visibility to complete invisibility.

Individual data were averaged across blocks for each observer to fit individual curves (Fig 6). As expected, at size 1.0 (i.e., 1.0 width and 1.0 height), the ellipse was easy to detect for all observers. This may be because our eye movement tolerance was set to 1.5˚, while the estimated blind spot size was on average 6.28˚ × 7.03˚ (Table 2), so a small eye movement could cause the object to be seen. This could also be indicated by the larger standard error (error bars on Fig 6) for larger probe objects: observers' answers varied more for the size range from 0.87 to 1.07, suggesting imperfectly stable fixation. Another reason could be that the "Scaling" validation procedure oversimplified the blind spot shape as a perfect ellipse. At sizes that were very close to the estimated blind spot, the elliptical objects could partially extend outside the true blind spot, leading to positive responses. This is supported by inspection of Fig 4A and 4C. The diamond boundaries drawn to join border points in those plots were already close to or even extended outside (upper right) the blind spot heat map boundaries, so an ellipse circumscribing the diamond would extend further into areas of visibility.

In general, the visibility value fell to 0% at around probe object size 0.70. The 0% visible size nonetheless differed considerably between observers (mean = 0.70, SD = 0.10, Range = [0.52, 0.86]), with a maximum value of around 0.87 (observer No. 2), and a minimum value between 0.50 and 0.60 (observer No. 5).

Overall, although the visibility rating could be influenced by individual differences, fluctuation in individual performance, errors caused by eye movements, and mistaken responses,

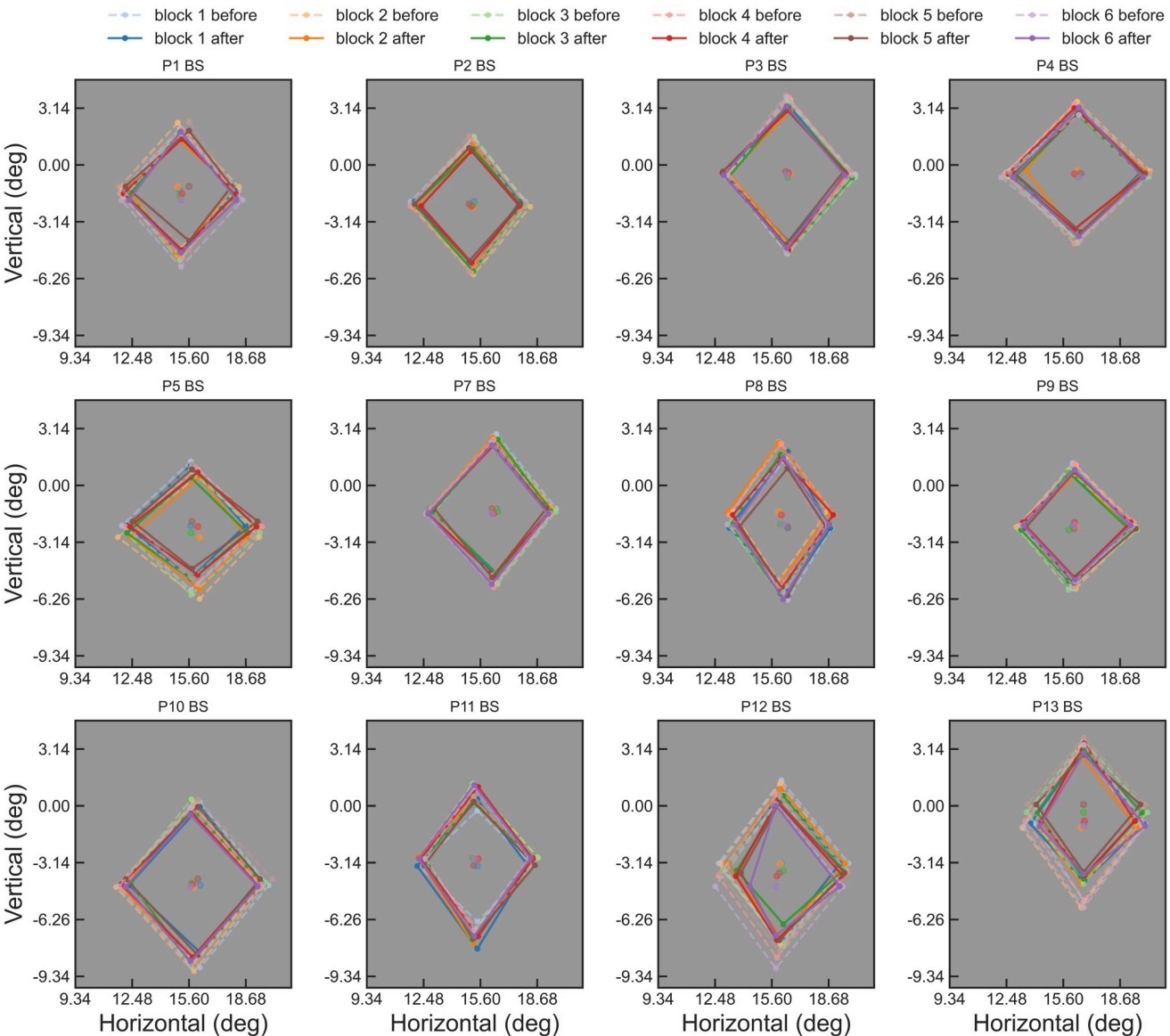

**Fig 5. Block-by-block blind spot location and border points estimates for 12 observers.** In the "Low" mode of the "Border points detection section", the cardinal border points were detected, so they form a diamond on this figure. The border points estimated before the Staircase section are joined with a dashed line, while those after the Staircase section are joined with a solid line. It is apparent that the blind spot location and size vary considerably between observers, and that the Staircase section has made the estimates more conservative. Note that the estimated blind spot location shifted vertically between blocks for observers No. 5, 8, 12, and 13.

0.50 could be taken as a very conservative scaling coefficient at which the elliptical object was likely to be completely invisible to all observers and randomly selected individuals (Fig 6).

The smallest estimated blind spot width and height were 5.39˚ and 5.91˚, respectively, so we propose that a 2.70˚ × 2.96˚ (0.5 size) region at the estimated blind spot center would be a "safe zone" within which any visual stimuli should be invisible to any observer. Thus, if researchers wish to confine visual stimuli safely within the blind spot, they just need to estimate the blind spot location and make their stimuli smaller than the safe zone. For this application, only the "Border points detection section" in "Low" mode would be required, while the "Staircase

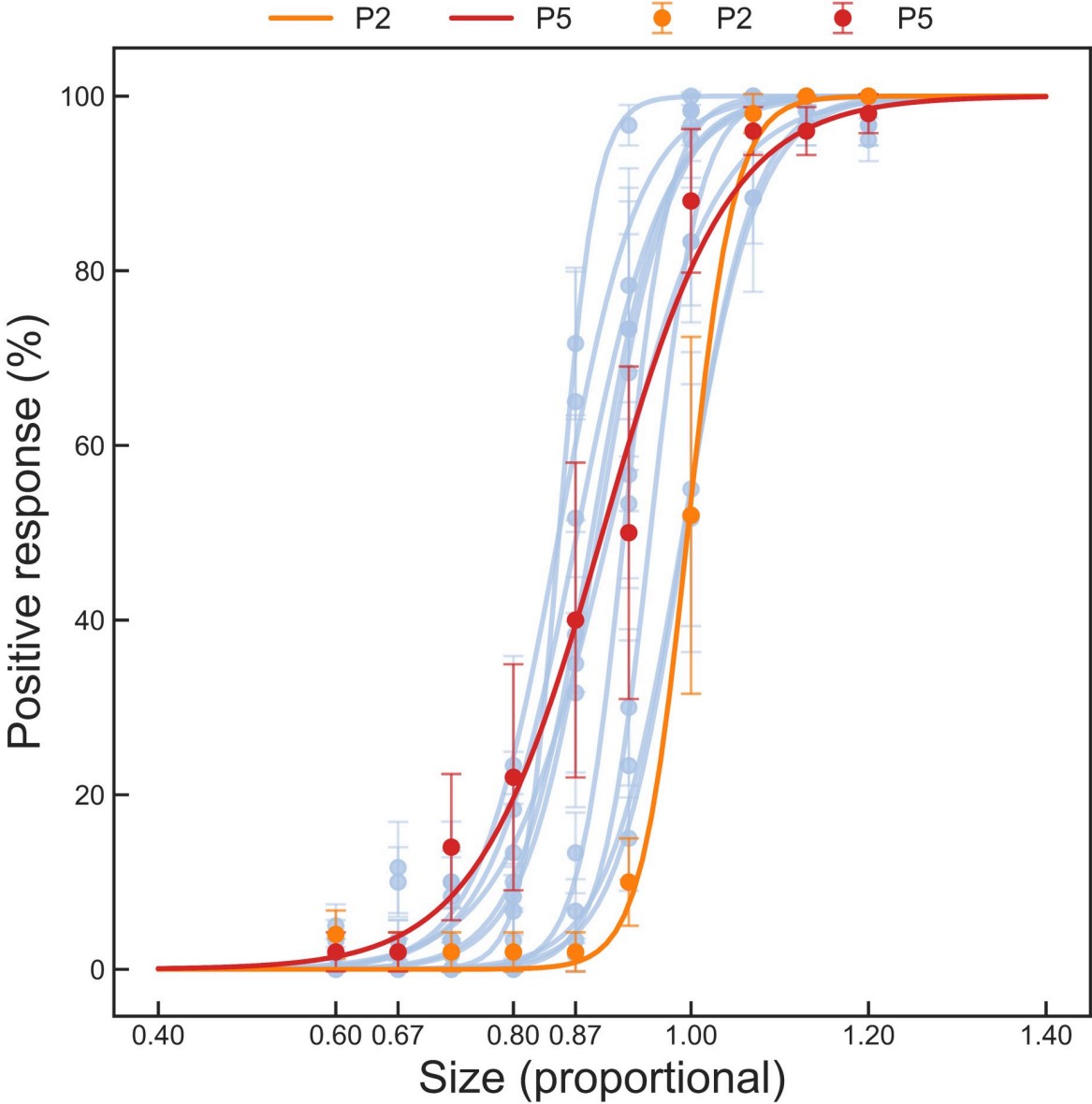

**Fig 6. Psychometric validation curves for 12 observers.** Within-observer data were averaged. Error bars indicate standard errors. For most observers, the visibility of probe objects decreased dramatically between sizes 1.00 and 0.80, and dropped to 0% at 0.67. Observers No.2 and 5 are highlighted to illustrate the total range of performance across observers, in terms of the smallest probe size at which positive responses first emerged.

section" and "Validation section" could be skipped entirely, providing a maximally efficient pragmatic mapping protocol for this important experimental application. In our Experiment 2, the "Border points detection section" was completed within 5 min in 49%, 8 min in 90%, and 14 min in 97% of 70 blocks, and in only two blocks, this section was completed in 19 and 21 min, respectively. These times include within-experiment breaks and the subsequent recalibration of the eye-tracker.

## Discussion

In this article, we have explained the rationale of our blind spot assessment tool and presented data collected using this tool. We also provided some practical suggestions on shortening the entire estimation block to optimize efficiency.

Experiment 1 generated two heat maps of the blind spot for one observer. These heat maps verified the blind spot shape, location, and size, as well as confirming the viability of the three-step method used in the Border points detection section of this tool. The results in Experiment 1 also indicated that the "Low" mode should be good enough to accurately estimate the blind spot size and location, while "Medium" mode could capture more border shape details.

Experiment 2 was conducted to evaluate the psychometric properties of this tool and to explore potential shortened routines for blind spot estimation. The location and size data revealed large individual differences, confirming that the blind spot must be mapped individually for experimental work. By testing a group of observers using "Low" mode estimation and "Scaling" validation, this tool appeared to be reliable and valid. The blind spot data (location and size) were in line with earlier studies (Table 1). A 0.7 proportional-size elliptical object located at the estimated blind spot location was invisible for most observers, and there was a $2.70˚ \times 2.96˚$ (0.5 scaling coefficient) "safe zone" at the estimated blind spot location within which the invisibility of a probe stimulus could be guaranteed. In fact, some studies have already used stimuli of similar sizes to hide them within the blind spot [45].

Based on these findings, we offer some tips for the use of our tool. First, "Low" mode is good enough to estimate the location and size of the blind spot, while "Medium" and "High" modes are more suitable to capture progressively more details of blind spot shape. Second, the Validation section can be omitted if researchers only want to estimate the blind spot location and size. Third, the Staircase section and the Validation sections can be skipped, leaving only the Border points detection section, if researchers just want to place visual stimuli (smaller than the safe zone) inside the blind spot. Finally, of course, it is worth emphasizing that the "Heat Map" procedure remains the gold-standard assessment for detailed mapping of the blind spot, where the precise shape is of interest, although this procedure is inefficient in terms of time.

In the interests of keeping the time-costs within reasonable bounds, some compromises were made in choosing the default settings for this tool. For example, the staircase rules were 20-trial, 4-reversal, and final-1-trial. The trial number of 20 is somewhat small. Computing the threshold based solely on the final trial is also not a common psychophysical practice; the more common way is to compute the mean of final-n reversals or trials. However, we had tested 50-trial, 5-reversal, and final-10-trial rules in a pilot study with 3 observers, who reported being impatient and frustrated, and strongly suggested reducing the number of trials to half of the original. Moreover, the staircase traces in that pilot study indicated that after around 10 to 15 trials, 3 reversals had been made, and the staircase had already reached the smallest step size. This pattern can also be seen in Fig 3. Hence, we decided to run a staircase in 20 trials in 4 reversals, so the staircase could run a small number of trials to reach the threshold level, and we arbitrarily chose the final-1 trial so that it must be after or at the 4th reversal.

The pilot study also helped us to determine the size of the flickering probe, which was used in the Border points detection section and the Staircase section, to make experimental blocks more smooth and easy to complete for observers. The relatively large eye movement tolerance (1.5˚ from the fixation position to the center of fixation object) was chosen for the same reason. Overall, we determined our default settings as pragmatic choices, which should provide high enough data quality, without being overly taxing for observers. These default settings can be adjusted in the graphical user interface, or by editing the configuration files or source code.

Finally, one unexpected observation is suggested by the heat maps shown in Fig 4. Conventionally, the blind spot is thought to be completely blind, however, the heat maps suggested that there was a small region inside the blind spot of that observer that was slightly sensitive to light. The heat maps showed that the observer responded positively (1 out of 10 trials) to the target presented within the blind spot (horizontal = 17˚, vertical = -2˚). This unexpected positive response is unlikely to have been any form of response errors, because this light-sensitive region was found independently in both heat maps at the same location (Fig 4, compare A and B with C and D), around 2˚ from the outer edge. This finding is surprising but consistent with earlier studies, which claimed that light inside the blind spot could induce a pupillary light reflex [38, 47]. However, since the sample size is a single observer, this finding needs to be further examined.

In summary, our blind spot assessment tool appeared to be reliable and valid. Although researchers have used similar procedures in previous studies, no freely accessible tool has been released, and the psychometric properties of previous methods lack validation. By contrast, data obtained by our tool have clear psychometrical meanings, and this tool has been validated. We hope it is a tool of value to vision science, and that it will be used by future researchers. The blind spot mapping tool, its user manual for researchers, and the full data reported in this manuscript, are available at https://github.com/LxIiNaGo/BlindSpotMapping.

## Author Contributions

**Conceptualization:** Xiao Ling, Edward H. Silson, Robert D. McIntosh.

**Data curation:** Xiao Ling.

**Formal analysis:** Xiao Ling.

**Investigation:** Xiao Ling.

**Methodology:** Xiao Ling.

**Software:** Xiao Ling.

**Supervision:** Edward H. Silson, Robert D. McIntosh.

**Validation:** Xiao Ling.

**Visualization:** Xiao Ling.

**Writing – original draft:** Xiao Ling.

**Writing – review & editing:** Xiao Ling, Edward H. Silson, Robert D. McIntosh.

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
