## [Decision Letter · Decision Letter 0]

29 Jul 2021

PONE-D-21-20867

Did you see it? A Python tool for psychophysical assessment of the human blind spot

PLOS ONE

Dear Dr. McIntosh,

Thank you for submitting your manuscript to PLOS ONE. As you will see from the comments below, both reviewers were positive about the paper but had some suggestions for improvement and comments that you will need to respond to before the paper can be accepted. Therefore, we invite you to submit a revised version of the manuscript that addresses the points raised during the review process.

We look forward to receiving your revised manuscript.

Kind regards,

Nicholas V Swindale

Academic Editor

PLOS ONE

Journal Requirements:

2. Please change "female” or "male" to "woman” or "man" as appropriate, when used as a noun (see for instance https://apastyle.apa.org/style-grammar-guidelines/bias-free-language/gender).

4.Please review your reference list to ensure that it is complete and correct. If you have cited papers that have been retracted, please include the rationale for doing so in the manuscript text, or remove these references and replace them with relevant current references. Any changes to the reference list should be mentioned in the rebuttal letter that accompanies your revised manuscript. If you need to cite a retracted article, indicate the article’s retracted status in the References list and also include a citation and full reference for the retraction notice.

Reviewers' comments:

Reviewer's Responses to Questions

**Comments to the Author**

1. Is the manuscript technically sound, and do the data support the conclusions?

Reviewer #1: Yes

Reviewer #2: Yes

2. Has the statistical analysis been performed appropriately and rigorously? 

Reviewer #1: Yes

Reviewer #2: Yes

3. Have the authors made all data underlying the findings in their manuscript fully available?

Reviewer #1: Yes

Reviewer #2: Yes

4. Is the manuscript presented in an intelligible fashion and written in standard English?

Reviewer #1: Yes

Reviewer #2: Yes

5. Review Comments to the Author

Reviewer #1: This paper provides a useful tool for standardising plotting of the blind spot. It is clearly and thoughtfully presented and easy to read. I do not have any major concerns about the paper.

I have a couple of minor concerns. The first refers to the window of tolerance used by the eye-tracker. If 1.5 deg tolerance was permitted, then the eye could move across a 3x3 deg region during data collection. That is almost half the width or height of the blind spot. I wish a more realistic window had been chosen. I would have liked to see data with a tolerance of 0.5 deg for the eye-tracker. I suspect that this will not make much of difference with regard to the measured size of the blind spot. However, it would have been a bit more reassuring, because if the eye is moving by as much as 3 deg, then the blind spot edge is moving by as much as 3 deg. The reason this does not show in the data is probably because, even though the eyes were permitted to move over a big window, the observers were sufficiently trained that they did not actually move their eyes. But if this tool were to be used widely and users were tempted to use this study as a basis for using a 1.5 deg window, there would occasionally be subjects (particularly when testing patients with visual disability) who would make large eye-movements during testing.

It would be helpful if some data were presented with a smaller tolerance during eye-tracking.

Additionally I think a few simulations would make the results interesting. Imagine a 7x6 deg oval blind spot - what would the measured size of the blind spot be if the eye was randomly positioned within a 0.5x0.5 deg window, 1.0x1.0 deg window, 1.5x1.5 deg window on each trial? I think this is an easy simulation to do. It would help us predict the consequences of poor fixation on mapped blind spot size.

The other concern I have has to do with potential mismatches between some numbers in the text part of the paper and those in the tables. It is possible I might have misunderstood something.

Last para on Pg 21 says:

"On average (bottom row of Table 2), the right blind spot was located at 16.00° temporal (right) to the fixation (SD = 0.52°, Range = [15.17°, 16.73°], n = 12), and slightly lower than the horizontal meridian (mean = - 2.05°, SD = 1.21°, Range = [-4.33°, -0.50°], n = 12)."

I can see the mean values match what I see in Table 2, but the SD and range do not seem to match. Perhaps I missed something?

First para on Pg 23 says:

"Overall, the blind spot width was 6.28° (SD = 0.62°, CV = 0.10, Range = [5.39°, 7.44°], n = 12), and the height was 7.02° (SD = 0.77°, CV = 0.11, Range = [5.79°, 8.14°], n = 12)."

Again the mean values match Table 2, but again the SD and range do not seem to match.

Reviewer #2: The manuscript by Ling, Silson, & McIntosh introduces a new software package to psychophysically measure the human retinal blind spot. This is a welcome addition to publicly available tools, since many studies need to measure the location of the retinal blind spot. In the past, many authors have implemented their own tools, but a publicly available tool will surely help to make research on visual scotomata and perceptual filling-in more accessible and more reproducible.

In the paper, the authors present the workflow of the software and the various options that were implemented to measure, correct, and verify blind spot boundaries. They go on and validate the tool in detailed measurements of a single subject’s blind spot, as well as with less detailed (yet more realistic) measurement parameters in a group of several subjects. This serves to show the usability and validity of the tool.

I want to commend the authors for providing such a valuable tool, and for presenting this useful assessment of its validity!

I would only ask the authors to address a few minor comments, which would help to clarify some issues.

1.) Figure 1B Caption:

“Note that in the 1st-horizontal-test of “Medium” mode, the separation between the middle and upper paths is the same as the middle-lower separation, and they are predetermined as 1.875°(7.5°/4, or 240 pix/4 when the viewing distance is 57 cm), which is much smaller than a half of a typical blind spot height. The separation between the 1st-vertical-test paths is not predetermined but is set to 1/4 of the estimated blind spot width obtained by the 1st-horizontal-test. The 2nd- horizontal-test has only one path. “

I failed to parse this text on first pass. I guess I don’t understand the placing of the horizontal test paths, and which order they are being run in. It would help to label these in the figure. A more logical procedure would be, after the bs-center is determined after the 2nd horizontal path, to have paths passing through the center of the BS, with additional test points placed equidistantly along the border. Maybe this is something to consider for a future release of the software.

2.) Page 17: The calculation of trial numbers is a little confusing. Maybe specify in further detail that the sum includes trial numbers for each phase of the experiment.

3.) Experiment 2: It would help to specify from the outset that multiple blocks are run for each observer to be able to assess with-in subject consistency.

4.) Caption of Figure 3: Maybe don’t label answers as Wrong and Correct, since technically they aren’t either.

5.) Table 1: Wang 2017 TVS?? or MVS

6. PLOS authors have the option to publish the peer review history of their article (what does this mean?). If published, this will include your full peer review and any attached files.

Reviewer #1: No

Reviewer #2: No

---

## [Author Response · Author response to Decision Letter 0]

2 Sep 2021

See Response to Reviewers document

---

## [Editor Report · Decision Letter 1]

8 Sep 2021

Did you see it? A Python tool for psychophysical assessment of the human blind spot

PONE-D-21-20867R1

Dear Dr. McIntosh,

We’re pleased to inform you that your manuscript has been judged scientifically suitable for publication and will be formally accepted for publication once it meets all outstanding technical requirements.

Kind regards,

Nicholas V Swindale

Academic Editor

PLOS ONE
---

## [Editor Report · Acceptance letter]

26 Oct 2021

PONE-D-21-20867R1 

Did you see it? A Python tool for psychophysical assessment of the human blind spot 

Dear Dr. McIntosh:

I'm pleased to inform you that your manuscript has been deemed suitable for publication in PLOS ONE. Congratulations! Your manuscript is now with our production department. 

Kind regards, 

on behalf of

Dr. Nicholas V Swindale 

Academic Editor

PLOS ONE